# Antibiotic Resistance in the Apennine Wolf (*Canis lupus italicus*): Implications for Wildlife and Human Health

**DOI:** 10.3390/antibiotics12060950

**Published:** 2023-05-23

**Authors:** Camilla Smoglica, Simone Angelucci, Fabrizia Di Tana, Antonio Antonucci, Fulvio Marsilio, Cristina Esmeralda Di Francesco

**Affiliations:** 1Post-Graduation School of Animal Health, Breeding and Zootechnical Productions, Department of Veterinary Medicine, University of Teramo, Loc. Piano D’Accio, 64100 Teramo, Italy; simone.angelucci@parcomajella.it (S.A.); fmarsilio@unite.it (F.M.); cedifrancesco@unite.it (C.E.D.F.); 2Wildlife Research Center, Maiella National Park, Viale del Vivaio, 65023 Caramanico Terme, Italy; fabrizia.ditana@parcomajella.it (F.D.T.); antonio.antonucci@parcomajella.it (A.A.)

**Keywords:** antibiotic resistance, wolf, wildlife, multidrug resistant bacteria, critically important antibiotics, conservation medicine, public health, one health

## Abstract

The Apennine wolf (*Canis lupus italicus*) is a subspecies of gray wolf that is widespread throughout Italy. Due to hunting and habitat loss, their population declined dramatically in the late 19th and early 20th centuries, but conservation efforts improved to restore the species to an estimated population of 3300 individuals. The presence of antibiotic-resistant bacteria in Apennine Wolf may pose a risk to its health and survival, as well as the health of other animals in its environment. In this study, we investigated the antibiotic resistance profiles of bacteria collected from Apennine wolves admitted to the Wildlife Research Center of Maiella National Park (Italy) in 2022. A total of 12 bacteria collected from four wolves were isolated and tested for susceptibility to antibiotics used in veterinary medicine and to critically important antibiotics for human health by means of the Vitek 2 system. All isolates were resistant to at least one antibiotic, and six bacteria were multidrug resistant to critically important antibiotics (third-generation cephalosporins, carbapenems and fluoroquinolones). The results of this pilot study have allowed for the characterization of resistant profiles in *Escherichia coli*, *Enterococcus faecalis* and other bacterial species not previously reported in Apennine wolves. Our findings provide important insights into antibiotic resistance in wildlife and its potential implications for the conservation of biodiversity and public health.

## 1. Introduction

The grey wolf *Canis lupus* (Linnaeus, 1758), a large carnivore of the *Canidae* family, is widely distributed throughout Eurasia and North America [1]. In Europe, at the end of the last century, the wolf survived in highly fragmented populations as a result of legal human persecution [2]. The Appennine wolf (*Canis lupus italicus*) is a subspecies of gray wolf that was once widespread throughout the Italian peninsula according to historical records [3]. However, due to massive hunting and habitat loss, the population of Apennine wolves declined dramatically in the late 19th and early 20th centuries, with highly fragmented populations located in Central and Southern Italy [4,5,6]. By the mid-20th century, the species was considered endangered and in need of protection [2]. In response to the declining population of Apennine wolves, conservation efforts were put in place in the 1970s and 1980s. This included the creation of protected areas and the implementation of hunting regulations [7]. The species is actually protected under the Bern Convention and the EU Habitats Directive, with the aim of conserving biodiversity and ensuring the survival of threatened species [8,9].

From the Central Apennines, wolves recolonized the entire Apennine chain and the Western Alps, where they established stable packs [10]. In detail, the Maiella National Park, located in Central Italy, has been one of the last areas where Apennine wolves survived human persecution, and its population worked as a source for the recolonization of the Apennine chain and Western Alps.

These efforts have been successful, and today, the population of Apennine wolves is estimated to be around 3300 individuals [11].

The Apennine wolf can be considered a keystone species, playing an important role in maintaining the balance of the ecosystem, and its conservation is essential for the health of the environment as a whole. [12]. However, the Apennine Wolf still faces challenges, such as habitat fragmentation, increased mortality due to illegal killing or road crossing and the loss of biodiversity, enhancing the probability of contact with domestic animals and anthropic activities. The abovementioned determinants have led to a higher risk of pathogens transmission or epidemic events, along with an increased number of injured wolves recovered from anthropic landscapes [13,14,15,16].

In this regard, the Apennine wolf, as an apex predator which may live in habitats with different ecological features and food availability, may be considered a sensitive indicator of antimicrobial resistance (AMR), one of the most important health threats for both humans and animals. Indeed, considering its environmental and trophic adaptation, this species can be exposed to several sources of resistant bacteria or antibiotic residues, harbored by the preys or humanized landscapes [17].

Antibiotic resistance is a growing concern in both human and animal health, and recent studies have suggested that wildlife may serve as sentinels of antibiotic resistance pollution in the environment [18,19,20,21,22]. Furthermore, the issue of antibiotic resistance in wildlife has important implications for conservation efforts. Indeed, the exposure of the Apennine wolf to antibiotic-resistant bacteria may pose a risk to its health and survival, as well as for other animals and the environment. The monitoring of antibiotic resistance in wildlife, including the Apennine wolf, is important for both human and animal health and for the conservation of biodiversity [19]. This approach is consistent with the One Health principles that promote the collaboration among various sectors, including human medicine, veterinary medicine, environmental science, public health and wildlife conservation, for maintaining healthy environments and functioning ecosystems.

The One Health approach recognized the AMR as a global problem considering the interconnections between humans, domestic animals and wildlife. A multidisciplinary strategy for combatting AMR is crucial for ensuring effective actions, preserving the efficacy of antibiotics and safeguarding the health of humans, animals and the environment [18,19].

In this view, the aim of the study was to report the antibiotic resistance profiles of bacteria isolated from injured Apennine wolves admitted to the Wildlife Research Center (WRC) of Maiella National Park (MNP) (Central Italy) in order to evaluate the role of the Apennine wolf as a sentinel species of AMR diffusion.

## 2. Results

During the year 2022, a total of four wolves were rescued alive and admitted to the veterinary facilities of the WRC for medical care and rehabilitation (Table 1).

In detail, animal LU251122VA was found in a very serious clinical condition, with severe hypothermia, cachexia and respiratory distress, and was in a comatose state. It had extensive skin disease suspected to be caused by mites, and it died after 5 days of clinical intensive care and treatments.

Animal LU120922 was captured for monitoring purposes and died after 3 days with sub-acute symptoms of unknown origin. Tachycardia and hyperthermia were observed during the capture, but blood tests showed parameters within the normal range [24,25,26].

It was treated with a trihydrate amoxicillin (150 mg/mL) long-acting single administration at 15 mg/kg.

Animal LU120522OUT was recovered with a lacerated wound from a metal snare illegal trap in the right carpal region with exposure of the radiocarpal joint. The animal was treated with enrofloxacin (50 mg/mL) at 5 mg/kg q12 h and clindamycin (600 mg/4 mL) at 10 mg/kg q12 h for 10 days. Based on the antibiotic susceptibility testing, the antibiotic therapy was replaced with cefovecin (80 mg/mL) at 8 mg/kg (single administration).

Animal LU270222AB was polytraumatized with a deep lacerated wound in the distal part of the right forearm caused by a metal snare trap, with radiographic evidence of osteomyelitis and bone remodeling involving the radius and ulna. A broad-spectrum therapy was administered, combining marbofloxacin (10 mg/mL) at 3 mg/kg q24 h and clindamycin (600 mg/4 mL) at 10 mg/kg q12 h (10 days of therapy). Based on the antibiotic susceptibility testing, the antibiotics were replaced with sulfadiazine/trimethoprim (200 mg/mL + 40 mg/mL) at 25 mg + 5 mg/kg q24 h (7 days of therapy).

A total of 12 samples were analyzed for bacterial isolation (8 tissues and exudates samples from deceased animals and 4 swabs from alive wolves). As reported in Table 2, thirteen bacteria were isolated, including five Gram-positive and eight Gram-negative strains.

All isolates were found to be resistant to at least one antibiotic, and six isolates were multidrug resistant (MDR) (resistant to at least three classes of antibiotics) (Figure 1).

## 3. Discussion

The results of this study provide valuable information on antibiotic resistance in the Apennine wolf. The current literature offers limited information on this topic, with studies showing the resistance profiles mainly derived from opportunistic fecal sampling [27,28,29]. Additionally, the available data are mostly related to indicator bacteria such as *Escherichia coli* and *Enterococcus* spp., without clear links to specific animals or their home ranges. In this study, the monitoring activities carried out by the MNP’s technical scientific staff made it possible to establish a correlation between the health outcomes and the ecological data of the animals involved.

*Escherichia coli* and enterococci isolated in this study show resistance to a wider range of antibiotics compared to the study carried out in the same protected area on domestic and wild ruminants [20,21,22]. Furthermore, the MDR *E. coli* isolates show resistance to critically important antibiotics (enrofloxacin, marbofloxacin, and pradofloxacin).

A similar pattern was also observed while analyzing *Enterococcus faecalis* strains, showing resistance to enrofloxacin and marbofloxacin. To the authors’ knowledge, this is the first evidence of resistance profiles against fluoroquinolones, which are considered antibiotics relevant for human medicine, in wolf species. Indeed, previous reports carried out on similar isolates coming from Italian and Iberian wolves are restricted to the resistance against tetracyclines, erythromycin and gentamicin [28,30].

Additionally, *Staphylococcus pseudintermedius*, *Klebsiella oxytoca*, *Pseudomonas aeruginosa*, *Streptococcus canis*, *Streptococcus dysgalactiae* subsp. *equisimilis* (SDSE) and *Leclercia adecarboxylata* were characterized for the first time in Apennine wolf.

*Staphylococcus pseudintermedius* is a pathogen that has been described in a variety of canine infections such as pyoderma and otitis or involving urinary, respiratory and reproductive tracts [31,32]. The resistance patterns highlighted in this study were associated with benzylpenicillin, amoxicillin and clavulanic acid, whereas in more recent reports, the methicillin-resistant strains appear to be of growing interest [31]. Therefore, it is important to collect data on this bacterial species in order to supervise the resistance trend of this bacterial species considered an emerging pathogen in humans [33].

Like *S. pseudintermedius*, the isolates of *K. oxytoca* that are resistant to ampicillin, along with SDSE, identified in this study are relevant. Both species have been recently recognized as emerging pathogens of animals and humans, with regard for nosocomial infections [32,34,35,36,37]. Notably, the SDSE strain isolated in this study showed multiple resistances to different classes of antibiotics.

The other MDR strains described are *S. canis*, *P. aeruginosa*, and *L. adecarboxylata*.

*Streptococcus canis* is considered a multi-host pathogen causing diseases of varying severity in a wide range of mammals, including humans [38]. *Pseudomonas aeruginosa* is a well-known opportunistic pathogen in veterinary medicine [32,39] and has recently been included in the World Health Organization’s list of priority pathogens [40]. *Leclercia adecarboxylata* is a rarely identified pathogen isolated from environmental and clinical specimens [41] which has gained interest in recent years due to its association with infections in immunocompromised human patients [42].

It is worth noting that all the MDR strains have one or more resistances to third-generation cephalosporins, carbapenems, fluoroquinolones and macrolides, which are defined by the World Health Organization as antibiotics of critical importance in human medicine [43]. These antibiotics are considered the last resource for the treatment of multi-resistant infections in humans.

Despite the small sample size, the data reported in this study highlight the importance of monitoring wild species to provide information on the environmental contamination by emerging or public health-relevant antibiotic resistance. Other authors have suggested that wolves, as apex predators, may have a resistance profile closely related to their trophic sources [29]. However, in a previous study, wolf packs with a similar diet showed different genetic resistance profiles [44]. Indeed, the diet of these wolves mainly constituted wild prey; nevertheless, their territories showed different degrees of anthropization [45]. This evidence suggested that the environment with different anthropic impacts may influence the resistance profiles of the animals rather than their hunting behavior, and its role should be further investigated. Therefore, it is essential to increase the number of samples coming from this species that may be considered a sentinel of ecosystem health [46].

Other free-ranging wild animals, such as the Apennine wolves involved in this study, have previously been recognized as indicators of antibiotic resistance contamination in the environment [17]. Specifically, free-ranging animals include wildlife with no history of domestication or exposure to animal husbandry practices. Furthermore, it has been hypothesized that omnivorous, anthropophilic and carnivorous species are at a higher risk of carrying and potentially spreading antimicrobial-resistant bacteria [17]. Recent scientific literature has provided evidence of the association between the occurrence of antibiotic-resistant bacteria in wildlife and their habitat use, their foraging strategies and human influences on the environment. However, the exact pathways of transmission between humans, domestic animals and wildlife have not yet been fully elucidated [18]. In this regard, investigations based on the One Health approach have been strongly recommended for identifying specific sources of antibiotic resistance in wildlife, assessing the associated risks for human and animal health and identifying effective management interventions [18].

To date, the literature data on antibiotic resistance associated with the activities of wildlife rehabilitation centers have been carried out in the USA, Chile, Spain and Italy, but they are mainly focused on wild birds, mustelids and ungulates [47,48,49,50,51,52,53]. Therefore, this study provides novel data on resistant bacteria in rescued wolves.

Collecting swabs and biological samples from animals found in emergency situations is an important process for understanding the health status of the animal and conserving the wildlife populations [54]. As suggested by previous studies [15], close physical contact between pack members is an important characteristic of social canids such as wolves, increasing the likelihood of pathogens transmission within the pack [55]. Therefore, the detection of pathogens or resistant bacteria in one or more samples suggests that multiple animals were likely exposed to the same agents [56]. By using laboratory investigations, researchers can better understand the health and population status of wildlife species and develop effective strategies for their protection and recovery.

The investigated animals have shown clinical conditions that have required an antibiotic treatment. The antibiotic susceptibility test was useful for improving the therapy, identifying the most appropriate molecules against the isolated bacteria. In addition, these data can be useful in enhancing a prudent use of antimicrobials to preserve the efficacy of the therapy used in wildlife medicine.

Finally, the antibiotic susceptibility profiles of bacteria can be useful not only in defining the most effective therapies for recovered animals but also in providing insights into public health considerations regarding environmental contamination by AMR.

## 4. Materials and Methods

The samples were recovered from Apennine wolves rescued alive and admitted to WRC veterinary facilities for medical care and rehabilitation during the 2022 year. Anamnestic data and clinical procedures including antibiotic therapy were registered from each animal. The medical procedures were carried out by the veterinary staff of WRC of MNP in Central Italy. The WRC mission is the protection of wildlife, livestock and people through ecological research, the surveillance of wildlife diseases and investigations into the wildlife–livestock interface. The multidisciplinary WRC team includes wildlife veterinarians, biologists, experts in animal ecology, wildlife management and forensic medicine and organic farming professionals. One of the WRC activities is the rescue and rehabilitation of sick, injured and orphaned native wildlife. This activity has significantly increased in recent years.

### 4.1. Samples Collection

The animals in distress were captured by chemical immobilization for clinical examination [57]. Sampling was conducted during these procedures without causing any disturbance to the animals. The animal capture procedures and clinical recovery were performed in accordance with the authorization from the Italian Institute for Environmental Protection and Research n.27895-23/08/2010 and Ministry of Environment n. 0018803-8/09/2010. This kind of authorization is required for the capture of the protected animals in the 92/43/EEC Directive and includes all guidelines that concern the procedures of capture, animal welfare and sanitary sampling.

Mucosal and cutaneous swabs were obtained from wounds of the alive animals by using sterile swabs and then placed into 5 mL of sterile tryptic soy broth (TSB) (Liofilchem^®^, Roseto degli Abruzzi, Italy).

Deceased wolves were submitted to necropsy in order to highlight any pathological gross lesions of organs. The post-mortem examination of deceased wolves revealed, for LU251122VA, a generalized lymphadenomegaly, moderate pleural effusion, suspected purulent pneumonia and the presence of thrombus obstructing both the left atrium and left ventricle, suggesting septic embolism. Animal LU120922 showed a large pleural effusion associated with suspected pneumonia, areas of pulmonary atelectasis, mild peritoneal effusion and macroscopic changes consistent with pancreatitis (pancreatic edema, congestion of the duodenal wall and hemorrhagic peritoneal congestion).

Tissues samples and exudates were collected from wolves during necropsy using sterile scalpels or syringes and then transferred into 5 mL of sterile TSB.

All samples were kept at +4 °C until the microbiological investigations.

### 4.2. Laboratory Investigations

The samples were aerobically incubated at 35 ± 2 °C for 18–24 h using the Chromatic^TM^ Detection Medium (Liofilchem^®^, Roseto degli Abruzzi, Italy) for the morphological identification of grown colonies, and the VITEK^®^ system (Biomerieux, Marcy-l’Étoile, France) was applied for the species identification of bacteria. In order to identify the therapeutic options, the antibiotic susceptibility tests were carried out using the VITEK^®^ system (Biomerieux) according to the guidelines of the European Committee on Antimicrobial Susceptibility Testing (EUCAST) [58].

Based on the bacterial strain identification, the panel of antimicrobial agents was used for antimicrobial susceptibility tests, including 26 drugs from 9 categories: aminoglycosides (amikacin, kanamycin, neomycin and gentamicin); penicillins (ampicillin, amoxicillin/clavulanic acid, benzylpenicillin and oxacillin); carbapenems (imipenem); first-generation cephalosporins (cephalothin and cephalexin); third-generation cephalosporins (ceftiofur, cefovecin and cefpodoxime); fluoroquinolones (enrofloxacin, pradofloxacin and marbofloxacin); tetracyclines (doxycycline, minocycline and tetracycline); macrolides (erythromycin); lincosamides (clindamycin); phenolics (florfenicol and chloramphenicol); nitrofurantoin and trimethoprim/sulfamethoxazole.

## 5. Conclusions

This report can be considered a pilot study that is useful in providing novel data on AMR patterns from bacterial strains recovered during the clinical care activities carried out on wolves rescued alive in Central Italy. The results obtained suggest the definition of a standardized surveillance system that is able to highlight any potential source of AMR dissemination in wildlife and the environment using the Apennine wolf as the sentinel species. Due to the challenge in determining the directionality of resistant bacteria transmission, it is crucial to conduct investigations and implement surveillance that involve wildlife as well as spatially and temporally related human and veterinary populations based on the One Health approach.

Without collective action, the deaths related to AMR could increase even further, reaching up to 10 million deaths annually by 2050 and causing a 3.8% reduction in the annual gross domestic product, with regard to people living in low- and middle-income countries [59].

The detection of bacteria resistant to critically important antibiotics in wildlife has previously emphasized the role of wild animals as indicators of environmental contamination [18]. These efforts are necessary for identifying potential trends and dissemination pathways of AMR [18].

## Figures and Tables

**Figure 1 antibiotics-12-00950-f001:**
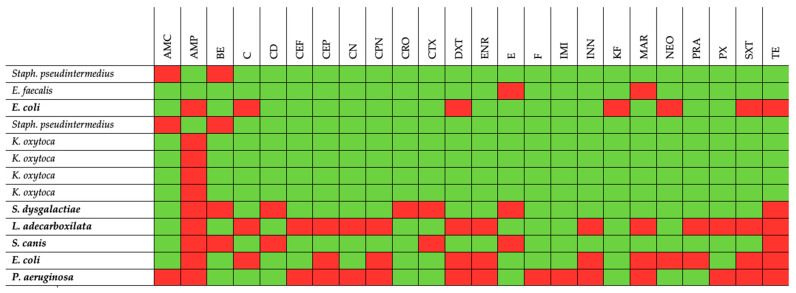
Phenotypic AMR profiles of bacterial isolates obtained from wolves under study. AMC: amoxicillin/clavulanic acid; AMP: ampicillin; BE: benzypenicillin; C: chloramphenicol; CD: clindamycin; CEF: ceftiofur; CEP: cephalothin; CN: gentamicin; CPN: cephalexin; CRO: ceftriaxone; CTX: cefotaxime; DXT: doxycycline; ENR: enrofloxacin; E: erythromycin; F: nitrofurantoin; IMI: imipenem; INN: cefovecin; KF: cephalothin; MAR: marbofloxacin; NEO: neomycin; PRA: pradofloxacin; PX: cefpodoxime; SXT: trimethoprim/sulfamethoxazole; TE: tetracycline; bold: multidrug resistant isolates; red: resistant; green: sensitive.

**Table 1 antibiotics-12-00950-t001:** Anamnestic data and samples recovered from wolves under study.

ID Animals	Sex	Age	BCS and Weight	Samples
LU251122VA	F	6 m	2/9 5.3 kg	Endocardium
				Lung
				Thoracic effusion
LU120922	F	2 y	4/9 25.4 kg	Peritoneal effusion
				Lung
				Endocardium
				Liver
				Pleural effusion
LU270222AB	F	7 y	3/9 22.9 kg	Forearm wound swab
				Exposed fracture swab
LU120522OUT	M	1 y	5/9 26.3 kg	Carpal wound swab
				Intraarticular swab

F: female; M: male; m: month; y: year; BCS: body condition score [23].

**Table 2 antibiotics-12-00950-t002:** Bacterial isolates and resistance profiles obtained from wolves under study.

ID Animals	Sample	Bacteria	Antibiotic Resistance Profiles
LU251122VA	Endocardial swab	*Staphylococcus pseudintermedius*	AMC BE
		*Enterococcus faecalis*	E * ENR * MAR *
	Lung	*Escherichia coli*	AMP DXT C KF NEO * SXT TE
	Thoracic effusion	*Staphylococcus pseudintermedius*	AMC BE
LU120922	Peritoneal effusion	*Klebsiella oxytoca*	AMP
	Lung	*Klebsiella oxytoca*	AMP
	Endocardial swab	*Klebsiella oxytoca*	AMP
	Liver parenchyma	*Klebsiella oxytoca*	AMP
	Pleural effusion	Negative	-
LU270222AB	Forearm wound swab	*Streptococcus dysgalactiae ssp equisimilis*	AMP BE CD CRO CTX E TE
	Exposed fracture swab	*Leclercia adecarboxilata*	AMP C CEF CEP CN CPN DXT ENR INN MAR PRA PX SXT TE
LU120522OUT	Carpal wound swab	*Streptococcus canis*	AMP BE CD CTX E TE
		*Escherichia coli*	AMP C CPN CEP DXT ENR INN MAR NEO * PRA SXT TE
	Intraarticular swab	*Pseudomonas aeruginosa*	AMC AMP CEF CEP CN CPN DXT ENR F IMI INN MAR * PX SXT TE

AMC: amoxicillin/clavulanic acid; AMP: ampicillin; BE: benzypenicillin; C: chloramphenicol; CD: clindamycin; CEF: ceftiofur; CEP: cephalothin; CN: gentamicin; CPN: cephalexin; CRO: ceftriaxone; CTX: cefotaxime; DXT: doxycycline; ENR: enrofloxacin; E: erythromycin; F: nitrofurantoin; IMI: imipenem; INN: cefovecin; KF: cephalothin; MAR: marbofloxacin; NEO: neomycin; PRA: pradofloxacin; PX: cefpodoxime; SXT: trimethoprim/sulfamethoxazole; TE: tetracycline; *: intermediate.

## Data Availability

The data presented in this study are available on request from the corresponding author.

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
