# Peer review of "Antibiotic Resistance in the Apennine Wolf (Canis lupus italicus): Implications for Wildlife and Human Health"

_antibiotics, 2023, doi:10.3390/antibiotics12060950_

Round 1
Reviewer 1 Report
General comments:
Manuscript entitled "Antibiotic resistance in the Apennine wolf: implications for wildlife and human health" describes study of bacterial resistance for many antibiotics in case of bacteria isolated from Apennine wolf. Living near human the species may be another natural indicator of presence of some bacteria found in the area common for wild animals and human therefore the monitoring process is crucial for our health. Study taken by the Authors have fundamental maening for recognition the status of antibiotic resistance in bacteria infecting wildlife and being threatened for human and domestic animals or livestock. Manuscript presents the pilot study but it points rightly on necessity of continuing featured work. The manuscript show the title problem briefly but the Authors do not avoid some failings.
I recommend the manuscript for publishing after necessary corrections.
Minor comments:
line 86 - the comma is needed in the end of the sentence.
line 184 - "the relevant key role" is to much, please, remove the word "relevant" or "key".
line 216 - there is "4.2", it should be "4.1".
line 228 - when the name of the producer is mentioned again the ful description is not required, for example "Biomerieux" is enough.
Author Response
Reviewer 1
General comments:
Manuscript entitled "Antibiotic resistance in the Apennine wolf: implications for wildlife and human health" describes study of bacterial resistance for many antibiotics in case of bacteria isolated from Apennine wolf. Living near human the species may be another natural indicator of presence of some bacteria found in the area common for wild animals and human therefore the monitoring process is crucial for our health. Study taken by the Authors have fundamental maening for recognition the status of antibiotic resistance in bacteria infecting wildlife and being threatened for human and domestic animals or livestock. Manuscript presents the pilot study but it points rightly on necessity of continuing featured work. The manuscript show the title problem briefly but the Authors do not avoid some failings.
I recommend the manuscript for publishing after necessary corrections.
We are very happy to have received a positive evaluation, and we would like to express our appreciation to Reviewer 1 for the thoughtful comments and helpful suggestions.
Our detailed, point-by-point responses to the reviewer comments are given below.
Minor comments:
line 86 - the comma is needed in the end of the sentence.
Done
line 184 - "the relevant key role" is to much, please, remove the word "relevant" or "key".
The text was modified following the suggestions of Reviewer 2 and Reviewer 3 removing the emphasis on “the key role.”
line 216 - there is "4.2", it should be "4.1".
Done
line 228 - when the name of the producer is mentioned again the ful description is not required, for example "Biomerieux" is enough.
Done
Reviewer 2 Report
This manuscript provides results from an antibiotic resistance survey in a wild species (the Apennine wolf). Studies like this provide useful information about antibiotic resistance and how big and broad this problem is. Therefore, the authors should receive some credit for their work.
Nevertheless, I think the manuscript needs to suffer considerable improvements before publication in its content and format. A deeper interpretation is needed. Here I present them, point-by-point, for the authors' consideration.
Title: The scientific name of the species should appear in the title after the common name.
Abstract
From my perspective, the abstract has a large introduction/background section and no methodology is mentioned. results or conclusions are also very briefly mentioned. Authors should reorganize the abstract providing similar information amount in each subsection, even if the abstract does not have subheadings.
Introduciton
L 49 : A definition of the ecology concept "keystone species" should be provided for the readers that are not familiar with this nomenclature.
L52-53 : Which contacts/consequences of contact are your refering to? Hunting? Diseases? I would rephrase this sentence a little bit a be more specific to what is threatening this species?
L57-58: I believe you should explain a little bit more why you think this wolf is a good bioindicator. why apex predators are good indicators? Trophic chain position? This should be a little bit more developed in my opinion. You should be careful because antibiotic residues or antibiotic resistance may not work as other contaminants with health importance.
Considering the end of your introduction and the importance of antibiotic resistance for humans, animals and the environment, the concept of One Health should be introduced and developed at the introduction.
Results
Although it is quite obvious, the authors should clarify what red and green colours mean in Table 3.
Discussion/Conclusion
I think both sections should be developed a little bit more. In the discussion, the authors should mention some limitations of the study. In both sections, the authors should provide a more developed reflection about what can be done in the future, considering this scientific field to prevent or to monitor with greater detail this One Health problem; what is the worse case predictive scenario for this situation....
Material and Methods
An ethical statement should be presented. For dead animals, even if is just for saying that no statement from an ethical board was necessary to conduct this study because these animals died naturally in rescue centres or were euthanised according to the rescue centre protocols, or something similar to this, but a reference should be made.
In alive animals, you need provide details regarding it (methods for restraint etc...). There is a lot of information missing in the process of sample collection.
I have nothing else to add. I wish the authors all the best.
Author Response
Reviewer 2
This manuscript provides results from an antibiotic resistance survey in a wild species (the Apennine wolf). Studies like this provide useful information about antibiotic resistance and how big and broad this problem is. Therefore, the authors should receive some credit for their work.
Nevertheless, I think the manuscript needs to suffer considerable improvements before publication in its content and format. A deeper interpretation is needed. Here I present them, point-by-point, for the authors' consideration.
We would like to express our appreciation to Reviewer 2 for the thoughtful comments and helpful suggestions.
Our detailed, point-by-point responses to the reviewer comments are given below.
Title: The scientific name of the species should appear in the title after the common name.
Done
Abstract
From my perspective, the abstract has a large introduction/background section and no methodology is mentioned. results or conclusions are also very briefly mentioned. Authors should reorganize the abstract providing similar information amount in each subsection, even if the abstract does not have subheadings.
The abstract has been improved in the way kindly suggested by Reviewer 2.
Introduciton
L 49 : A definition of the ecology concept "keystone species" should be provided for the readers that are not familiar with this nomenclature.
Done
L52-53 : Which contacts/consequences of contact are your refering to? Hunting? Diseases? I would rephrase this sentence a little bit a be more specific to what is threatening this species?
In agreement with the Reviewer 2, the causes of threat have been specified (Lines 52-57).
L57-58: I believe you should explain a little bit more why you think this wolf is a good bioindicator. why apex predators are good indicators? Trophic chain position? This should be a little bit more developed in my opinion. You should be careful because antibiotic residues or antibiotic resistance may not work as other contaminants with health importance.
The text has been modified based on the Reviewer’s suggestions (Lines 58-63)
Considering the end of your introduction and the importance of antibiotic resistance for humans, animals and the environment, the concept of One Health should be introduced and developed at the introduction.
The introduction has been improved in the way kindly suggested by Reviewer 2 (Lines 71-79).
Results
Although it is quite obvious, the authors should clarify what red and green colours mean in Table 3.
Done
Discussion/Conclusion
I think both sections should be developed a little bit more. In the discussion, the authors should mention some limitations of the study. In both sections, the authors should provide a more developed reflection about what can be done in the future, considering this scientific field to prevent or to monitor with greater detail this One Health problem; what is the worse case predictive scenario for this situation....
The Discussion (Lines 191-204) and Conclusions(Lines 276-292) have been improved in the way kindly suggested by Reviewer 2 and Reviewer 3.
Material and Methods
An ethical statement should be presented. For dead animals, even if is just for saying that no statement from an ethical board was necessary to conduct this study because these animals died naturally in rescue centres or were euthanised according to the rescue centre protocols, or something similar to this, but a reference should be made.
In alive animals, you need provide details regarding it (methods for restraint etc...). There is a lot of information missing in the process of sample collection.
Materials and Methods have been improved in the way kindly suggested by Reviewer 2 and Reviewer 3. (Lines 235-256).
Reviewer 3 Report
An interesting manuscript, in general worthy of publication. I have few comments for improvement of the final version.
Introduction. Please define clearly the objectives of the manuscript.
M & M. Please explain the procedures for getting the four cadavers for post-examination and sampling, as well as the procedures for catching the live animals for sampling.
Results.
Some of the details of the four animals that died, would fit better in M&M.
The presentation of the findings from dead and live animals is not correct.
Discussion. The topic merits a longer discussion really, especially to present the sylvatic cycle of transmission of resistant bacteria.
Moderate editing of English language.
Author Response
Reviewer 3
An interesting manuscript, in general worthy of publication. I have few comments for improvement of the final version.
We would like to express our appreciation to Reviewer 3 for the thoughtful comments and helpful suggestions.
Our detailed, point-by-point responses to the reviewer comments are given below.
Introduction. Please define clearly the objectives of the manuscript.
The introduction has been improved in the way kindly suggested by Reviewer 2 and the aim of the study was clarify in lines 80-83.
M & M. Please explain the procedures for getting the four cadavers for post-examination and sampling, as well as the procedures for catching the live animals for sampling.
Materials and Methods have been improved in the way kindly suggested by Reviewer 2 and Reviewer 3 (Lines 235-256).
Results.
Some of the details of the four animals that died, would fit better in M&M.
The presentation of the findings from dead and live animals is not correct.
Results have been modified as suggested by Reviewer 3. The details concerning the necropsy carried out on deceased animals were moved to M&M.
Discussion. The topic merits a longer discussion really, especially to present the sylvatic cycle of transmission of resistant bacteria.
The Discussion (Lines 192-204) and Conclusions(Lines 276-292) have been improved in the way kindly suggested by Reviewer 2 and Reviewer 3.
Round 2
Reviewer 2 Report
Tha authors have sufficiently improve their manuscript according to my suggestions and corrections.
Author Response
Thank you very much for your review report. We really appreciate your comments and suggestions.
Best regards
Reviewer 3 Report
The clinical consequences of these findings must be presented in a separate paragraph in the discussion.
Moderate editing of English language.
Author Response
Reviewer 3
The clinical consequences of these findings must be presented in a separate paragraph in the discussion.
The Discussion has been improved in lines 218-222 as suggested by the Reviewer.
Thank you for your comments and suggestions.